# Structural and Physiochemical Properties of Polyvinyl Alcohol–Succinoglycan Biodegradable Films

**DOI:** 10.3390/polym16131783

**Published:** 2024-06-24

**Authors:** Jae-pil Jeong, Inwoo Yoon, Kyungho Kim, Seunho Jung

**Affiliations:** 1Department of Bioscience and Biotechnology, Konkuk University, Seoul 05029, Republic of Korea; bruce171525@gmail.com (J.-p.J.); rudgh971225@naver.com (K.K.); 2Department of System Biotechnology, Konkuk University, Seoul 05029, Republic of Korea; dbsdlsdn123@konkuk.ac.kr

**Keywords:** succinoglycan, polyvinyl alcohol, biodegradable film, UV-blocking effect

## Abstract

Polyvinyl alcohol (PVA)–bacterial succinoglycan (SG) biodegradable films were developed through a solvent-casting method. Effects of the PVA/SG ratio on the thickness, transmittance, water holding capacity, and structural and mechanical properties were investigated by various analytical methods. All the prepared films were transparent and uniform, and XRD and FTIR analyses confirmed that PVA was successfully incorporated into SG. The films also showed excellent UV-blocking ability: up to close to 80% with increasing SG concentration. The formation of effective intermolecular interactions between these polymers was evidenced by their high tensile strength and moisture transport capacity. By measuring the biodegradation rate, it was confirmed that films with high SG content showed the fastest biodegradation rate over 5 days. These results confirm that PVA/SG films are eco-friendly, with both excellent biodegradability and effective UV-blocking ability, suggesting the possibility of industrial applications as a packaging material in various fields in the future.

## 1. Introduction

Petroleum-based polymers have been primarily used in packaging applications in recent years because of their excellent mechanical and barrier properties, easy processability, and relatively low cost. Despite these advantages, the carbon emissions generated during the production of petroleum-based polymers and their low biodegradability are major obstacles to the future continued use of these materials. Due to serious environmental concerns and consumer health, biopolymers or biodegradable plastics have attracted much research attention [1,2]. Biopolymers or biodegradable plastics can be classified as natural, synthetic, or derived from microbial sources: (1) natural-based polymers can be divided into polysaccharides, proteins, or lipids from animal and plant origins such as starch, cellulose, alginate, chitosan, soy protein, collagen, etc.; (2) synthetic-based polymers include poly(l-lactide) (PLA), poly(ε-caprolactone) (PCL), poly(vinyl alcohol) (PVA); (3) biopolymers extracted by microbial sources include poly(hydroxyalkanoates) (PHAs) and exopolysaccharides (EPSs) such as xanthan and curdlan [3]. Among these substances, polysaccharides are one of the most abundant substances in nature and can be extracted from natural resources such as animals, plants, and microorganisms. Polysaccharides have their own biodegradable, biocompatible, and non-immunogenic properties [4,5,6].

EPSs have been reported to have excellent rheological properties and potential applications in several fields, including the food, cosmetic, and pharmaceutical industries [7]. Based on these properties, they are recognized as candidates for use as coagulants, binders, emulsifiers, gelling agents, stabilizers, film formers, lubricants, and thickeners [8]. Succinoglycan (SG) is an EPS extracted from soil microorganisms such as *Agrobacterium* and *Sinorhizobium* species. Its structural composition consists of one galactose and seven glucose repeating units with chemical substituents such as acetyl, pyruvyl, and succinyl. SG is used in a wide range of fields, including the food, oil, and cosmetic industries, due to its emulsifying effect, high thermal stability, and viscosity [9]. Currently, SG is been expanding its applications into the biomedical field owing to its antibacterial and antioxidative effects [10]. Microbial EPS has useful properties such as high viscosity, economic efficiency, and short production cycle, but its application has limitations because of its low mechanical strength, heat transfer, barrier properties, and flexibility compared with petroleum-based polymers [11,12].

To overcome these limitations, researchers have tried to blend EPSs with plasticizers [13,14], proteins [15,16], nanoparticles [17,18], or other biopolymers [19,20]. To enhance the mechanical properties of EPS-based films, polyvinyl alcohol (PVA) is usually chosen as the matrix because of its excellent film-forming properties, biocompatibility, and degradability. PVA is a widely used water-soluble and biodegradable synthetic polymer with many hydroxyl groups in its molecular chain. Based on its mechanical and biological properties, it has applications in food packaging, biology, medicine, and construction [21]. In recent studies, PVA/EPS-based films have been reported in the biomedical and bioplastics fields [22,23,24,25,26]. However, there is currently no study investigating the basic properties of PVA/SG blended films.

This study aims to investigate the functionality of a film composed of a mixed solution of SG and PVA. Film-forming solutions containing PVA, SG, and glycerol were prepared via the solvent-casting method. Morphological, physical, and mechanical properties were analyzed using SEM, FTIR, XRD, and TGA analyses. Additionally, the moisture content, water solubility, and biodegradability of the films were investigated to determine the potential of biodegradable films for a wider range of applications.

## 2. Materials and Methods

### 2.1. Materials

The microbial strain (*Sinorhizobium meliloti* Rm1021) was provided by Konkuk University (Seoul, Republic of Korea). Glycerol (ACS reagent, ≥99.5%) and polyvinyl alcohol (PVA) (Mw ~ 89,000–98,000) were obtained from Sigma-Aldrich (Steinheim, Germany).

### 2.2. Extraction and Purification of SG

SG was extracted and purified as previously described [27]. *Sinorhizobium meliloti* Rm1021 was cultivated in media with glutamate–mannitol–salt (GMS) at 30 °C and pH 7 for 7 days. The supernatant and precipitate were separated from the cultivated media by centrifugation after 7 days, and the volume of the supernatant was reduced through evaporation. Pure ethanol was added three times to the evaporated supernatant to precipitate SG. The unpurified SG was redissolved in distilled water, dialyzed for 5 days, and lyophilized.

### 2.3. Preparation of PVA/SG Composite Films

PVA/SG composite films were fabricated via the solvent-casting method as described with slight modifications (Figure 1) [28]. Firstly, 2 g of SG was dissolved in 100 mL of distilled water at room temperature, and 2 g of PVA powder was dissolved in 100 mL of distilled water at 95 °C. After being fully dissolved, the PVA solution was cooled to room temperature. PVA/SG film-forming solutions were prepared by mixing each solution with glycerol (30 wt %) to a final volume of 100 mL (Table 1). The film-forming solution (15 mL) was poured into a petri dish (d ~ 10 cm) and dried at 55 °C for 24 h. The films were detached from the petri dishes and stored at room temperature within a desiccator.

### 2.4. Characterization

#### 2.4.1. Light Transmission Measurements

Light transmittance through the films (4 cm × 3 cm) was measured in the range of 200–800 nm on an ultraviolet–visible (UV-vis) spectrophotometer (UV-2450, Shimadzu, Kyoto, Japan). The UV-blocking properties and visible-light transmittance were evaluated using the transmittance at 280 nm (T_280_) and 600 nm (T_600_), respectively [29].

The opacity was calculated using the following equation [30]:Opacity = A_600_/x,(1)
where A_600_ is the absorbance of the film at 600 nm, and x is the mean thickness of the film (mm).

#### 2.4.2. FTIR

FTIR spectra of the PVA/SG films were measured by an FTIR spectrometer (TENSOR27, Bruker, Germany). The wavenumber range of 4000–650 cm^−1^ was covered ata a resolution of 1 cm^−1^.

#### 2.4.3. Thermogravimetric Analysis (TGA)

The thermal stability of PVA and PS films was evaluated by thermogravimetric analyzer (NETZSCH STA449F3, Selb, Germany). Measurement of the films was performed at a temperature with the range from 20 to 700 °C, a heating rate of 10 °C/min, and a N_2_ flow rate of 50 mL/min.

#### 2.4.4. X-ray Diffraction (XRD) Measurements

The crystal structures of the PVA/SG films were subjected to XRD using an X-ray analytical instrument (Rigaku SmartLab, Akishima, Japan). XRD patterns within the range of 2θ = 10–80° were measured at a tube voltage of 30 kV and a tube current of 20 mA under CuKα conditions.

#### 2.4.5. Field-Emission Scanning Electron Microscopy (FESEM)

Morphology observation and thickness measurements of the PVA and PS films were performed with FESEM images (JSM-7800 F, JEOL Ltd., Tokyo, Japan). PVA/SG films were cut (1 cm × 1 cm) and coated with platinum for 60 s at 20 mA in vacuum to endow them with electrical conductivity. Then, the films were observed at a magnification of 1500 times with an acceleration voltage of 5.0 kV.

#### 2.4.6. Mechanical Properties

The tensile strength (TS) and elongation at break (EB) of the films (2 cm × 6 cm) were measured using an Instron (E3000LT) micro-fatigue tester. The original grip distance was 40 mm, and the crosshead speed was 10 mm/s [31].

### 2.5. Moisture Content (MC) and Water Solubility (WS) Measurement

The moisture content (MC) and water solubility (WS) of the films were measured using slightly modified methods based on [32,33]. MC was measured by cutting out 1 × 1 cm^2^ rectangles from the circular films, which were dried at 105 °C for 24 h and then weighed.

MC was calculated using Equation (2):(2)$$\mathrm{MC}=\frac{\left(W_{0}-W_{1}\right)}{W_{0}}\times 100$$
where $W_{0}$ is the initial weight of the sample (g), and $W_{1}$ represents the weight after 24 h in the oven (g).

WS was measured by cutting out 1 × 1 cm^2^ rectangles from the circular films, which were dried at 105 °C for 24 h and then weighed. The cut films were then immersed in 10 mL of distilled water with stirring at room temperature for 24 h. The undissolved film was dried at 105 °C for 24 h and then weighed.

WS was calculated using Equation (3):(3)$$\mathrm{WS}=\frac{\left(W_{0}-W_{1}\right)}{W_{0}}\times 100$$
where $W_{0}$ is the weight (g) before immersion in distilled water, and $W_{1}$ is the weight (g) of the undissolved film after drying.

### 2.6. Biodegradability Test

Biodegradability tests were performed according to the following method to evaluate the biodegradability of the film [34]. A circular film with a diameter of 10 cm was placed 10 cm below the soil, and changes in the shape of the film were observed at intervals of 1–2 days.

## 3. Results

### 3.1. Optical Properties of PVA/SG Films

Photographs of the films are shown in Figure 2. The blended films were fabricated via the solvent-casting method [35]. Adding glycerol as a plasticizer to the film solution not only increased the elasticity, flexibility, and toughness of the film, but also improved the handling properties of the film by reducing brittleness and preventing cracks [36]. All the films were colorless and transparent (Figure 1). Also, the films were found to be smooth and flexible. As the SG concentration increased, the apparent transparency of the film decreased slightly.

To investigate the opacity and transmittance of the films, the UV absorbance and transmittance of the films were measured. As shown in Figure 3, the visual transmittance of the films slightly decreased as the SG concentration increased. Interestingly, the transmittance of the UVB region (T _280_) decreased significantly from 80.5 ± 0.32 to 20.1 ± 0.45 (Table 2), suggesting that SG could improve the UV-blocking effect of the films without adding any UV-blocking agent. According to previous reports, films composed of PVA/cellulose nanocrystals demonstrated a UV-blocking effect approximating 40% of the film. Additionally, a PVA/chitosan-based film showed a UV-blocking effect of approximately 60%. Compared with the results of other reports, the PVA/SG film was found to have a better UV-blocking effect compared with PVA/polysaccharide films without added UV-blocking additives [37,38]. UV light is one of the major causes of lipid oxidation; thus, PVA/SG films with good UV-blocking properties are suitable candidates for food packaging applications [39].

### 3.2. Characterization of the PVA/SG Films

#### 3.2.1. FTIR Analysis

FTIR spectra of the PVA/SG films are shown in Figure 4. PVA powder and purified SG were analyzed as control groups to compare the original functional properties of PVA and SG. PVA exhibited inherent characteristic peaks at 3302 cm^−1^ (–OH stretching vibration), 2951 cm^−1^ (asymmetric –CH_2_ stretching vibration), 1430 cm^−1^ (–C=O stretching vibration), 1162 cm^−1^ (PVA crystalline-sequence peak), and 1089 cm^−1^ (C–O–C stretching vibration) [40]. The characteristic peaks of SG were found at 3330 cm^−1^ (–OH stretching vibration), 2912 cm^−1^ (asymmetric –CH2 stretching vibration), 1723 cm^−1^ (C=O stretching vibration of acetyl ester), 1606 cm^−1^ (COO– asymmetric stretching vibration of pyruvate and succinic acid), 1368 cm^−1^ (COO– symmetric stretching vibration of pyruvate and succinic acid), and 1036 cm^−1^ (C–O–C stretching vibration) [41]. For PVA/SG films, the crystal peaks at 1162 cm^−1^ and 1089 cm^−1^ disappeared as the ratio of SG increased. The morphological changes in the PVA structure were explained by analyzing these peak areas [42]. Additionally, peaks were observed at 1723 cm^−1^ and 1606 cm^−1^, which represented the stretching vibrations of acetyl ester, succinic acid, and pyruvate, depending on the SG concentration. It was revealed that SG and PVA polymers were successfully blended within the film network.

#### 3.2.2. TGA and DTG Analysis

TGA and DTG analysis were conducted to check for the thermal stability of the PVA/SG films (Figure 5). For PVA/SG films, the first degradation step occurred when a small weight loss (13%) was observed at the temperature about 30–260 °C due to the loss of physically bound water molecules and the decomposition of volatile compounds [43]. The second maximum weight loss occurred between 240–450 °C (67%) due to the chain decomposition of PVA and SG [44]. After adding the SG polymer, the second degradation temperature was slightly increased by about 40–50 °C compared with that of the PVA film, suggesting that thermal stability of the film was improved by the intermolecular interaction between the PVA and SG polymers. Also, this result was consistent with previous reports that had reported SG as a potential material for improving thermal stability [45,46].

#### 3.2.3. XRD Analysis

In Figure 6, the XRD patterns of the PVA and PVA/SG films are shown. PVA was reported to have a semi-crystalline structure. Its characteristic peaks appeared at 2θ = 11.4°, 19.6°, 22.8°, and 40.9°, indicating the inherent patterns of the PVA crystalline structure [47,48]. The peak around 19.5° of SG, which had a broad pattern compared with PVA, revealed the structural amorphous nature of SG [49]. From the crystal pattern of the fabricated PVA film, it could be observed that the characteristic crystalline peak became weak. In the case of PVA/SG films, the peak intensities around 19.5° tended to decrease, and the peak at 40.9° disappeared as the amount of SG increased. This could be explained by the increase in intermolecular interactions between SG and the PVA polymer, resulting in a decrease in PVA crystallinity.

#### 3.2.4. FE-SEM Analysis

FE-SEM analysis was performed to measure the thickness of the films and to check for morphological changes. In Figure 7, the PVA film appeared to have a uniform and smooth cross-section. Its thickness was calculated to be 22.7 ± 0.18 μm. The thickness of all the PVA/SG films increased from 32.3 ± 0.48 μm to 36.6 ± 0.57 μm compared with the pure PVA films. Also, the morphology of PVA/SG films became uneven and rough compared with the PVA film. In the case of PS 3 and PS 4, some pores were observed in the cross-sectional images. Similar results were reported for the correlation between pores and polysaccharide concentration in the film cross-section. It had been reported that the uniformity and mechanical strength of the film tended to decrease when pores formed in the cross-section [50,51].

#### 3.2.5. Mechanical Properties

Figure 8 shows the stress–strain curves, tensile strength, and elongation at break values of the PVA/SG films. The PVA film had an average tensile strength of about 28.5 ± 3.0 MPa. When SG was blended in, the PS 1 and PS 2 films had higher values of tensile strength of about 38.5 ± 1.4 and 30.24 ± 0.02 MPa, respectively, than the PVA film. PS 3 and PS 4 films had lower values of tensile strength corresponding to 28.4 ± 1.0 and 18.7 ± 1.1 MPa. The cause of this phenomenon might be the pore formation and cross-sectional unevenness mentioned in the SEM above. Elongation of break values of the PVA and PVA/SG films gradually decreased as the ratio of SG increased. This could be explained by the motion of the polymer chain being limited by hydrogen bonds and the dispersed SG polymer. The decrease in EB values after SG addition might be due to the improved stiffness of the films [52].

#### 3.2.6. Moisture Content (MC) and Water Solubility (WS) of the Films

The moisture content (MC) values of the films can provide reasonable information to describe the interaction of PVA and SG with water molecules [53]. As shown in Figure 9A, there was a progressive rise in the MC value as the SG polymer concentration increased. The pure PVA film had the lowest MC value because PVA polymers can form compact hydrogen bonding with themselves, which interrupts their interaction with water molecules. When adding SG, the PVA and SG polymers formed hydrogen bonds with each other; so, a greater abundance of free PVA and SG polymers could interact more with water molecules. A similar phenomenon was reported about the interaction of a single polymer being weakened by an added polymer, resulting in an increase in the MC value [54,55].

The value of the water solubility (WS) of films appeared to follow the same trend as the MC. (Figure 9B) The PVA film was also measured as having the lowest WS value among the films. This trend could be associated with the establishment of hydrogen bonds between SG and PVA, which reduces the free hydroxyl groups in the film and hinders the interaction of water molecules and the polymer chains [56,57]. The biodegradation rate, one of the important properties of films, is often influenced by the hydrophilicity of the polymer matrix [58]. Since SG is known to be an exopolysaccharide extracted from soil microorganisms, it is less likely to cause environmental contamination in the soil. Therefore, based on the high moisture content and water solubility of the PVA/SG films, biodegradation testing of the films was performed in the soil environment.

### 3.3. Biodegradation Test of the Films

Figure 10 presents the results of the biodegradability testing of the films in the soil environment. In the case of the pure PVA film, its original shape was maintained even after 7 days. On the other hand, films with increased SG content were successfully degraded with a loss of film shape. The PS 4 film showed a tendency for its quality to deteriorate quickly within 5 days. When comparing the biodegradation rate with other PVA/polysaccharide films that were allowed to decompose for more than a week, the biodegradation rate of PVA/SG film was found to be fast [59,60]. The longer the blended film stayed in the soil, the more water it absorbed from the soil, helping it to be decomposed by soil microorganisms. Previous reports explained that the main factor affecting the decomposition rate of biodegradable films was known to be their ability to be decomposed by fungi and bacteria [61,62].

## 4. Conclusions

In this study, UV-blocking and biodegradable PVA/SG films were successfully manufactured. Structural, physicochemical, and morphological analyses demonstrated that the PVA/SG network was successfully integrated to form intermolecular interactions within the blended films. PVA/SG films showed effectively improved thermal stability compared with pure PVA films. The incorporation of SG through hydrogen bonding with PVA within PVA/SG films improved both the mechanical properties and the water holding capacity. Additionally, PVA/SG films were shown to exhibit relatively fast biodegradation rates compared with pure PVA films due to their high moisture capacity. In conclusion, the newly produced PVA/SG films are eco-friendly due to their rapid biodegradability and efficient UV-blocking effects, suggesting their potential as a new biomaterial in various film industries, including as packaging materials.

## Figures and Tables

**Figure 1 polymers-16-01783-f001:**
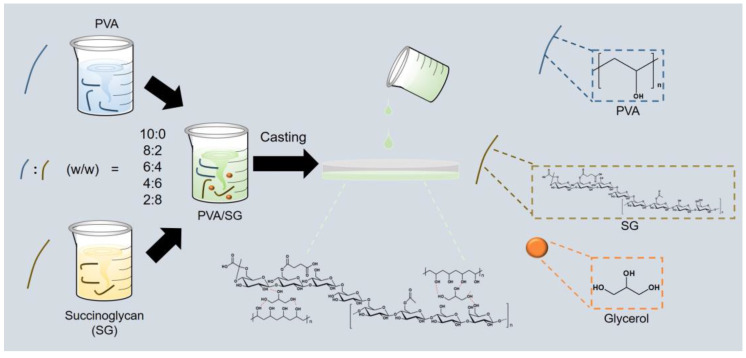
Scheme for preparing the PVA/SG films.

**Figure 2 polymers-16-01783-f002:**
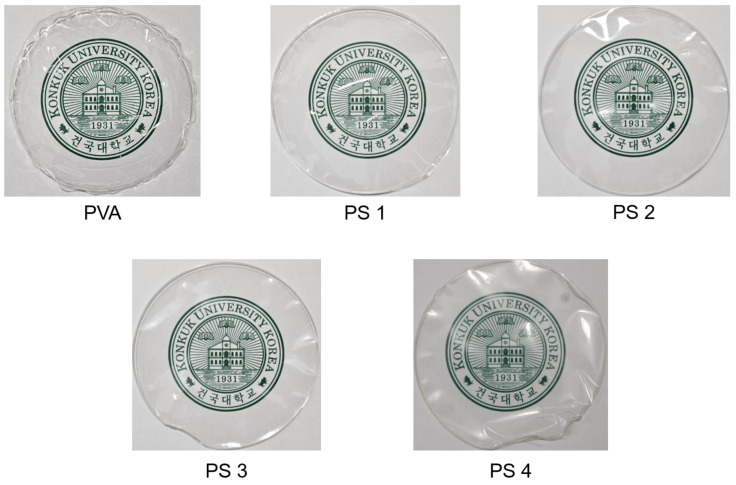
Photographs of the PVA/SG films.

**Figure 3 polymers-16-01783-f003:**
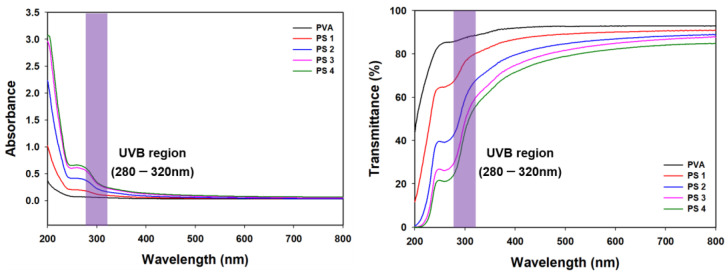
UV absorbance and transmittance of the films.

**Figure 4 polymers-16-01783-f004:**
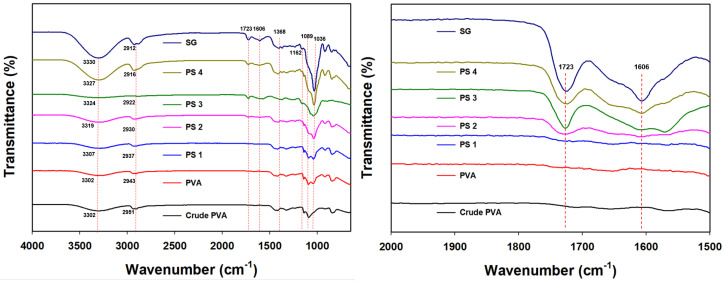
FTIR spectra of the PVA/SG films.

**Figure 5 polymers-16-01783-f005:**
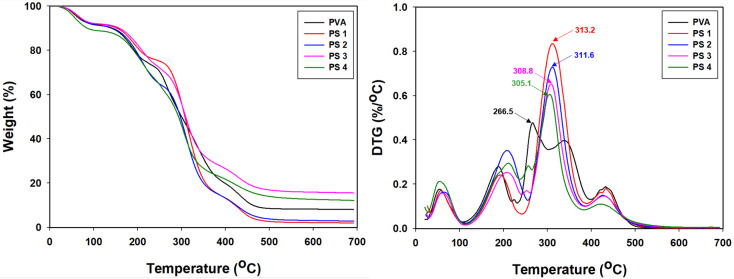
TGA and DTG curves of the PVA/SG films.

**Figure 6 polymers-16-01783-f006:**
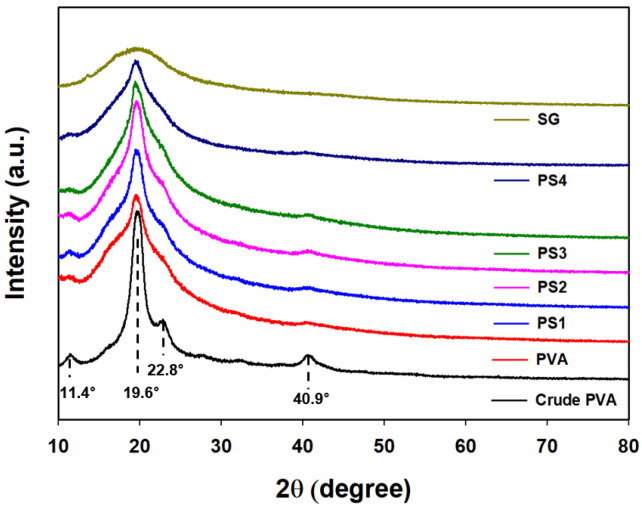
XRD pattern of the PVA/SG films.

**Figure 7 polymers-16-01783-f007:**
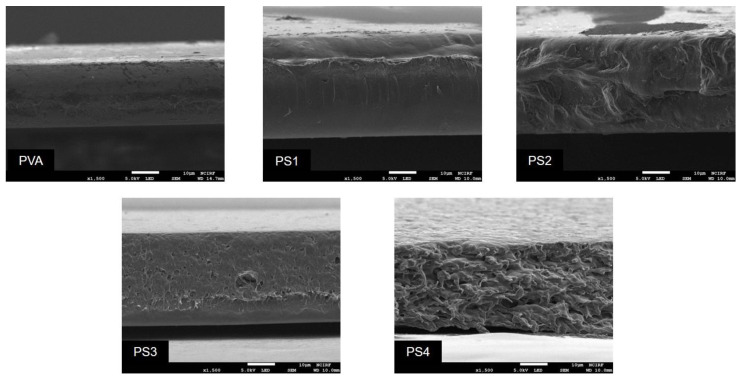
Cross-sectional SEM images of the PVA and PS films (×1500).

**Figure 8 polymers-16-01783-f008:**
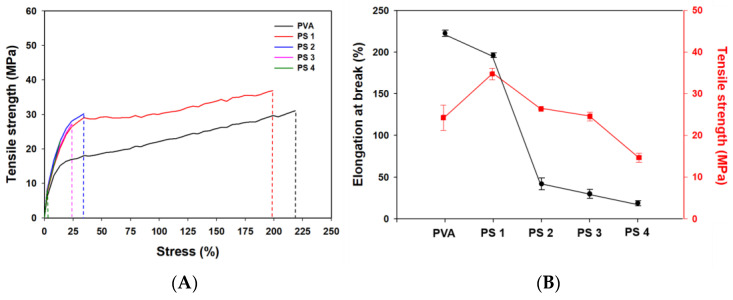
Stress-strain (**A**)and elongation at break (**B**) curves relative to tensile strength for the different films.

**Figure 9 polymers-16-01783-f009:**
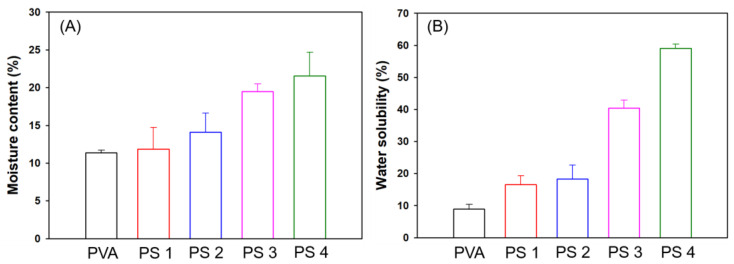
MC (**A**) and WS (**B**) of the films.

**Figure 10 polymers-16-01783-f010:**
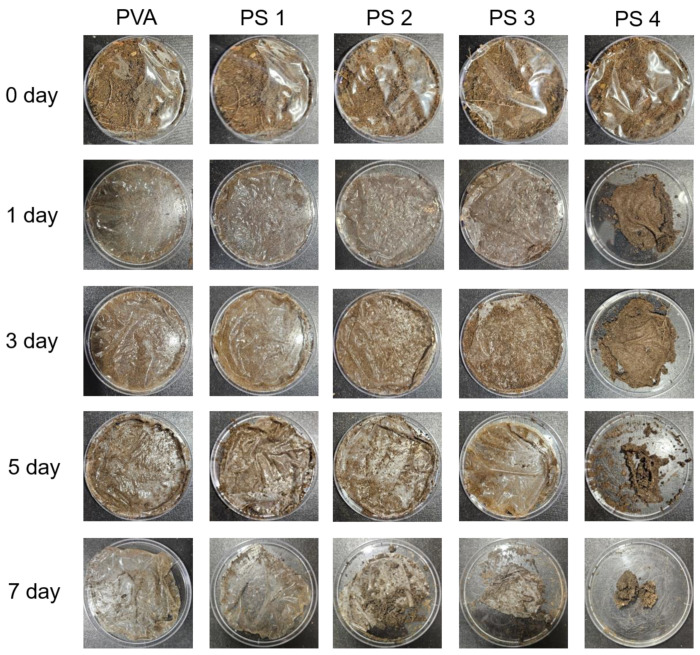
The biodegradability test of the films in a soil environment.

**Table 1 polymers-16-01783-t001:** Composition of PVA/SG films.

Film Name	PVA	SG	Glycerol	Water
PVA	2 g	-	30 wt %	100 mL
PS 1	1.6 g	0.4 g		
PS 2	1.2 g	0.8 g		
PS 3	0.8 g	1.2 g		
PS 4	0.4 g	1.6 g		

**Table 2 polymers-16-01783-t002:** Opacity and transmittance value of the films.

Film Name	Thickness (μm)	Opacity	T _280_	T _600_
PVA	22.7 ± 0.18	1.53 ± 0.13	80.5 ± 0.32	89.67 ± 0.75
PS 1	36.6 ± 0.57	1.18 ± 0.29	73.6 ± 0.52	89.2 ± 0.59
PS 2	34.7 ± 0.36	1.38 ± 0.42	52.6 ± 0.65	86.6 ± 0.72
PS 3	32.3 ± 0.48	2.03 ± 0.31	26.6 ± 0.57	73.3 ± 0.53
PS 4	35.1 ± 1.86	2.53 ± 0.85	20.1 ± 0.45	68.1 ± 0.42

## Data Availability

The original contributions presented in the study are included in the article, further inquiries can be directed to the corresponding author.

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
