# Peer review of "Structural and Physiochemical Properties of Polyvinyl Alcohol–Succinoglycan Biodegradable Films"

_polymers, 2024, doi:10.3390/polym16131783_

Round 1

Reviewer 1 Report

Comments and Suggestions for Authors

The experimental article “Structural and physiochemical properties of polyvinyl alcohol-succinoglycan biodegradable films” is devoted to the preparation and description of the properties of a promising material in the form of transparent films, which consists of polyvinyl alcohol (PVA)-bacterial succinoglycan (SG). By all criteria, the article corresponds to the Polymers publication. The positive aspects include a fairly detailed and very reliable study of the properties of the promising PVA/SG material, good quality illustrations (excellent Figures 1 and 9), and a constructive discussion of the results obtained. But there are also disadvantages. The main thing is that authors should work on reducing plagiarism. The reviewer checked the reason for plagiarism: 38% is associated with the Materials and Methods section, as well as the use of correct terminology in the field of polymers and methods of their research, which “responds” to any sources of information when searching in databases.

Specific recommendations:

1. It is recommended to reduce 33% of plagiarism.

2. In the introduction, it is recommended to indicate whether SG-based film formulations with excellent UV blocking effect are known.

3. The results must be presented in accordance with the sequence of methods described in the Materials and Methods section. The reviewer suggests changing the order of listing methods to implement this principle, so first in section “2.4. Characterization" will be "2.4.4 Light transmission measurements".

4. If the main purpose of new polymer films is “UV-blocking”, then it is recommended to devote more text to this property and provide comparisons with known solutions to prove the superiority of films based on PVA/SG.

5. Figure 9. It is recommended to provide quantitative results to assess biodegradability. The drawing looks really great, but clarity alone is not enough for such serious conclusions.

6. It is recommended to check reference 5, the year of publication is missing.

Author Response

The experimental article “Structural and physiochemical properties of polyvinyl alcohol-succinoglycan biodegradable films” is devoted to the preparation and description of the properties of a promising material in the form of transparent films, which consists of polyvinyl alcohol (PVA)-bacterial succinoglycan (SG). By all criteria, the article corresponds to the Polymers publication. The positive aspects include a fairly detailed and very reliable study of the properties of the promising PVA/SG material, good quality illustrations (excellent Figures 1 and 9), and a constructive discussion of the results obtained. But there are also disadvantages. The main thing is that authors should work on reducing plagiarism. The reviewer checked the reason for plagiarism: 38% is associated with the Materials and Methods section, as well as the use of correct terminology in the field of polymers and methods of their research, which “responds” to any sources of information when searching in databases.

Specific recommendations:

  1. It is recommended to reduce 33% of plagiarism.

 -> As your comment, we reduced plagiarism rate in the materials and methods section of the manuscript. As a result of checking through the plagiarism prevention program, the total plagiarism rate of the papers was 29%, However, after examining each paper, the plagiarism rate was confirmed to be less than 7%.

  1. In the introduction, it is recommended to indicate whether SG-based film formulations with excellent UV blocking effect are known.

-> Actually, we have searched about the previous reports of SG-based film with excellent UV blocking, however, there is no report about the investigation of property for PVA/SG films as we suggested in the manuscript. Therefore, our manuscript is the first report on SG-based film formulations with excellent UV protection effect.

  1. The results must be presented in accordance with the sequence of methods described in the Materials and Methods section. The reviewer suggests changing the order of listing methods to implement this principle, so first in section “2.4. Characterization" will be "2.4.4 Light transmission measurements".

-> As your comment, we reorganized the sequence of methods in the revised manuscript.

Before > 2.4.4 Light transmission measurements

Light transmittance through the films (4 cm x 3 cm) was measured in the range of 200-800 nm on an ultraviolet-visible (UV-vis) spectrophotometer (UV-2450, Shimadzu, Kyoto, Japan). The UV blocking properties and visible light transmittance were evaluated using the transmittance at 280 nm (T280) and 600 nm (T600), respectively [29].

The opacity was calculated using the following equation [30]:

Opacity = A600/x,  (1)

A600 is absorbance of the film at 600 nm, x is mean thickness of the film (mm).

After > 2.4.1 Light transmission measurements

Light transmittance through the films (4 cm x 3 cm) was measured in the range of 200-800 nm on an ultraviolet-visible (UV-vis) spectrophotometer (UV-2450, Shimadzu, Kyoto, Japan). The UV blocking properties and visible light transmittance were evaluated using the transmittance at 280 nm (T280) and 600 nm (T600), respectively [29].

The opacity was calculated using the following equation [30]:

Opacity = A600/x,  (1)

A600 is absorbance of the film at 600 nm, x is mean thickness of the film (mm).

  1. If the main purpose of new polymer films is “UV-blocking”, then it is recommended to devote more text to this property and provide comparisons with known solutions to prove the superiority of films based on PVA/SG.

-> As your comment, we added some more explanation and references of the UV-blocking effect compared to other PVA/polysaccharide film in the revised manuscript.

Before > To investigate the opacity and transmittance of films, UV absorbance and transmittance of the films were measured. As shown in Figure 3, The visual transmittance of the films was slightly decreased as SG concentration increased. Interestingly, the transmittance of UVB region (T 280) decreased significantly from 80.5 ± 0.32 to 20.1 ± 0.45 (Table 2), suggesting that SG could improve the UV-blocking effect of the films without adding any UV-blocking agent.

After > To investigate the opacity and transmittance of films, UV absorbance and transmittance of the films were measured. As shown in Figure 3, The visual transmittance of the films was slightly decreased as SG concentration increased. Interestingly, the transmittance of UVB region (T 280) decreased significantly from 80.5 ± 0.32 to 20.1 ± 0.45 (Table 2), suggesting that SG could improve the UV-blocking effect of the films without adding any UV-blocking agent. According to previous reports, films composed of PVA/cellulose nanocrystals observed a UV blocking effect of approximately 40% of the film. Additionally, the PVA/chitosan-based film showed a UV blocking effect of approximately 60%. Compared with the results of other reports, PVA/SG film was found to have better UV blocking effect compared to PVA/polysaccharide film without added UV blocking additives. [38,39].

[38] Yang, Weijun, et al. "Effect of cellulose nanocrystals and lignin nanoparticles on mechanical, antioxidant and water vapour barrier properties of glutaraldehyde crosslinked PVA films." Polymers 12.6 (2020): 1364.

[39] Liu, Fengsong, et al. "Improved hydrophobicity, antibacterial and mechanical properties of polyvinyl alcohol/quaternary chitosan composite films for antibacterial packaging." Carbohydrate Polymers 312 (2023): 120755.

  1. Figure 9. It is recommended to provide quantitative results to assess biodegradability. The drawing looks really great, but clarity alone is not enough for such serious conclusions.

->It is very difficult to completely remove the soil attached to the film, which makes it difficult to provide quantitative data on the soil biodegradability of the film.Therefore, we actually performed a biodegradability test on the film following the method presented in the following references (Zhang, Shaokai, et al., Carbohydrate Polymers 321 (2023): 121290; Xia, Qinqin, et al., Nature Sustainability 4.7 (2021): 627-635).

[3] Zhang, Shaokai, et al. "Fabrication of biodegradable films with UV-blocking and high-strength properties from spent coffee grounds." Carbohydrate Polymers 321 (2023): 121290.

[4] Xia, Qinqin, et al. "A strong, biodegradable and recyclable lignocellulosic bioplastic." Nature Sustainability 4.7 (2021): 627-635.

  1. It is recommended to check reference 5, the year of publication is missing.

-> As your comment, we checked the year of publication of reference 5.

Yadav, H.; Karthikeyan, C. Natural polysaccharides: Structural features and properties. In Polysaccharide carriers for drug delivery; Elsevier: 2019; pp. 1-17.

Thank you very much for your critical and useful comments.

Reviewer 2 Report

Comments and Suggestions for Authors

The authors have reported on the structural and physiochemical properties of polyvinyl alcohol-succinoglycan biodegradable films. The manuscript is very well-written and concise. However, the following comments need to be addressed:

  1. The authors have not mentioned the initial coating thickness or the drying mechanism used.
  2. The effect of the initial or final coating thickness on the degradation and mechanical properties has not been discussed.
  3. It is unclear whether the film will degrade in a moist environment without soil, or if soil bacteria are necessary for degradation.
  4. Although the title references structural changes, no structural formulas are provided in the manuscript.
  5. The ideal conditions for degradation have not been specified.
  6. Also address the minor changes marked in the attached file.

Author Response

The authors have reported on the structural and physiochemical properties of polyvinyl alcohol-succinoglycan biodegradable films. The manuscript is very well-written and concise. However, the following comments need to be addressed:

  1. The authors have not mentioned the initial coating thickness or the drying mechanism used.

-> We have mentioned about the drying mechanism in the method section and thickness of films was measured using FE-SEM analysis as the followings in the revised manuscript. We actually prepared the film using a drying mechanism called the solvent casting methods (Kumar, Himanshu, et al. Food and Bioprocess Technology 16.2 (2023): 342-355; Zeng, Shulong, Li Li, and Qi Wang. Polymer Testing 126 (2023): 108143).

[1] Kumar, Himanshu, et al. "Antioxidant film based on chitosan and tulsi essential oil for food packaging." Food and Bioprocess Technology 16.2 (2023): 342-355.

[2] Zeng, Shulong, Li Li, and Qi Wang. "Structure-property correlation of polyvinyl alcohol films fabricated by different processing methods." Polymer Testing 126 (2023): 108143

2.3. Preparation of PVA/SG composite films

PVA/SG composite films were fabricated via solvent casting method as described with slight modification [28]. Firstly, 2 g of SG was dissolved in 100 ml of distilled water at room temperature and 2 g of PVA powder was dissolved in 100ml of distilled water at 95 °C. After fully dissolved, PVA solution was cooled at the room temperature. PVA/SG film-forming solutions were prepared by mixing each solution with the final volume 100 ml (Table 1) with the glycerol (30 wt %). The film-forming solution (15 mL) was poured to a petri dish (d ~ 10 cm) and dried at 55 °C for 24 h. The films were removed from the petri dishes and stored at 25 °C in desiccator.

2.4.5 Field Emission Scanning Electron Microscopy (FE-SEM)

Morphology observation and thickness measurements of PVA and PS films were measured with FE-SEM (JSM-7800 F Prime, JEOL Ltd., Tokyo, Japan). The films were cut (1 cm x 1 cm) and coated with platinum for 60 seconds at 20 mA in vacuum to endow electrical conductivity. Then, the films were observed at a magnification of 1500 times with an acceleration voltage of 5.0 kV.   

  1. The effect of the initial or final coating thickness on the degradation and mechanical properties has not been discussed.

-> As a result of measuring the tensile strength of PS 1 throughout the study, PS 1 was found to have the highest tensile strength among the films, and PS 4 was found to have the fastest biodegradation rate. As reported in some papers (Zhang, Guangjie, et al. Journal of Food Science 86.2 (2021): 434-442; Vichasilp, Chaluntorn, et al. Food biophysics 9 (2014): 238-248) it generally shows that film thickness, mechanical properties and degradation rate are not related to each other.

[3] Zhang, Guangjie, et al. "Theoretical and experimental investigation of sodium alginate composite films containing star anise ethanol extract/hydroxypropyl‐β‐cyclodextrin inclusion complex." Journal of Food Science 86.2 (2021): 434-442.

[4] Vichasilp, Chaluntorn, et al. "Effect of longan seed extract and BHT on physical and chemical properties of gelatin based film." Food biophysics 9 (2014): 238-248.

  1. It is unclear whether the film will degrade in a moist environment without soil, or if soil bacteria are necessary for degradation.

-> The film can degrade in a moist environment without soil based on the result of water solubility test. Biodegradability test of films have performed in the soil because water and microorganism such as bacteria, fungi existed in soil could attack the polymer and degraded the film (Zhang, Shaokai, et al. Carbohydrate Polymers 321 (2023): 121290; Xia, Qinqin, et al. Nature Sustainability 4.7 (2021): 627-635). Also, succinoglycan can be extracted from the soil bacteria known as sinorhizobium or agrobacterium spp, we conducted the experiment of biodegradation in the soil environment.

[5] Zhang, Shaokai, et al. "Fabrication of biodegradable films with UV-blocking and high-strength properties from spent coffee grounds." Carbohydrate Polymers 321 (2023): 121290.

[6] Xia, Qinqin, et al. "A strong, biodegradable and recyclable lignocellulosic bioplastic." Nature Sustainability 4.7 (2021): 627-635.

  1. Although the title references structural changes, no structural formulas are provided in the manuscript.

-> Since the title of the manuscript is ‘Structural and physiochemical properties of polyvinyl alcohol-succinoglycan biodegradable films’, we provided the structural properties, the structural and composition changes of the films throughout the FTIR. XRD, TGA and FESEM analyses in the manuscript. We added some more discussions and references in the revised manuscript.

  1. The ideal conditions for degradation have not been specified.

 -> Accordingly, a biodegradation experiment was conducted in soil conditions at a depth of 5 to 10 cm under the same conditions as those presented in the reference (Zhang, Shaokai, et al., Carbohydrate Polymers 321 (2023) : 121290 ; Xia, Qinqin, et al. Nature Sustainability 4.7 (2021): 627-635).

[5] Zhang, Shaokai, et al. "Fabrication of biodegradable films with UV-blocking and high-strength properties from spent coffee grounds." Carbohydrate Polymers 321 (2023): 121290.

[6] Xia, Qinqin, et al. "A strong, biodegradable and recyclable lignocellulosic bioplastic." Nature Sustainability 4.7 (2021): 627-635.

  1. Also address the minor changes marked in the attached file.

-> As your comment, we corrected the minor changes marked in the attached file in the revised manuscript.

Thank you very much for your critical and useful comments.
